# TEMPORALLY-WEIGHTED SPIKE ENCODING FOR EVENT-BASED OBJECT DETECTION AND CLASSIFICATION

## ABSTRACT

Event-based cameras exhibit high dynamic range and temporal precision that could make them ideal for detecting objects with high speeds and low relative luminance. These properties have made event-based cameras especially interesting for use in space domain awareness tasks, such as detecting dim, artificial satellites with high brightness backgrounds using ground-based optical sensors; however, the asynchronous nature of event-based data presents new challenges to performing objection detection. While spiking neural networks (SNNs) have been shown to naturally complement the asynchronous and binary properties of event-based data, they also present a number of challenges in their training, such as the spike vanishing problem and the large number of timesteps required for maximizing classification and detection accuracy. Furthermore, the extremely high sampling rate of event-based sensors and the density of noisy space-based data collections can result in excessively large event streams within a short window of recording. We present a temporally-weighted spike encoding that greatly reduces the number of spikes derived from an event-based data stream, enabling the training of larger SNNs with fewer timesteps for maximal accuracy. We propose using this spike encoding with a variant of convolutional SNN trained utilizing surrogate spiking neuron gradients with backpropagation-through-time (BPTT) for both classification and object detection tasks with an emphasis on space-domain awareness. To demonstrate the efficacy of our encoding and SNN approach, we present competitive classification accuracies on benchmark datasets N-MNIST (99.7%), DVS-CIFAR10 (74.0%), and N-Caltech101 (72.8%), as well as state-of-the-art object detection performance on event-based, satellite collections.

## 1 INTRODUCTION

In recent years, the number of resident space objects (RSOs) in low-Earth orbit (LEO) and geosychronous-Earth orbit (GEO) has steadily grown, and consequently driven greater interest in the detection and tracking of such targets using ground-based optical telescopes. The tracking of RSOs, such as satellites or space debris, presents a unique challenge in that these targets often have very few distinguishing features from their surroundings and are difficult to image at high speeds. Furthermore, such targets are often far dimmer than ambient lighting, especially in both cis-lunar orbits and daytime viewing. These challenges motivate the need for new hardware sensors and computer vision techniques that can be easily integrated with existing ground-based detection schemes.

Event-based cameras, or dynamic vision sensors, are one attractive technology that presents a solution to imaging RSOs. These cameras operate without a global clock, allowing each individual pixel to asynchronously emit events based on detected changes in illuminance at high frequency. Each pixel exhibits a logarithmic response to illuminance changes, resulting in such cameras having large dynamic range. Furthermore, since pixels respond only to changes in illuminance, the data produced is far sparser compared to that of a conventional sensor sampling at comparable rates. Of perhaps crucial importance for space-based detection tasks, the operation of event-based pixels also prevents them from saturating, which could prove incredibly useful for imaging near the Moon or in daylight These qualities suggest that event-based cameras could be ideal for the detection of dim, high-speed RSOs that generally are too challenging for conventional CCD sensors.

However, the asynchronous nature of event-based data also poses a challenge to performing object detection effectively and efficiently. A naive approach to working with event-based data is to integrate over a pre-defined window of time to produce conventional images. Since each pixel generates events asynchronously, events are given both an $(x, y)$ location as well as a timestamp $t$ that corresponds to the time of event generation relative to a recorded starting time. Events are also given a polarity flag, $p \in \{1, -1\}$, that denotes the event was generated by an increase in illuminance (+1) or a decrease in illuminance (-1). For integration, these events, of the form $e = (x, y, t, p)$, are accumulated over some window $\Delta t$ at their respective $(x, y)$ locations to form an equivalent image. Such integrated frames can then be used with any conventional object detection method; however, this approach loses much of the temporal information present in the original event stream.

Spiking neural networks (SNNs) differ from conventional neural networks in much the same way that event-based data differs from conventional images. SNNs function asynchronously, with each neuron of the network generating spikes only when its inputs cause it to exceed a pre-defined threshold, mimicking the function of biological neurons. The sparsity of SNN activation amounts to spiking networks being exceptionally energy efficient as compared to conventional neural networks of comparable size. However, the binary nature of spiking neuron output and the subsequent non-differentiability of their activation makes supervised training of such networks a challenging task. Furthermore, SNNs are also plagued by the vanishing spike propagation issue, where decreasing spiking activity in successive layers causes significant performance loss in larger networks (Panda et al. (2020)). Nonetheless, the unique properties of SNNs naturally complement the data produced by event-based cameras, and multiple works have already shown the potential for classification and object detection on event-based data.

In this work, we present a temporal-weight encoding that greatly decreases the number of spikes derived from event data stream while maintaining overall spiking behavior and preserving temporal information. This encoding scheme is also shown to reduce the number of timesteps required to maximize classification and object detection accuracy in spiking neural networks. We also propose a pseudo-spiking behavior for conventional, convolutional neural networks that removes the need for temporal credit assignment, but preserves some temporal information. This pseudo-spiking behavior is readily integrated with encoded, event-based data and enables the training of comparatively deeper models than true spiking networks. We evaluate detection results using both simulated and real space-based data collection and demonstrate competitive performance on publicly available event-based classification and object detection datasets.

## 2  RELATED WORK

The following sections briefly explore some of the most notable works in each area touched upon in our own work.

### 2.1  SPACE DOMAIN AWARENESS

As previously mentioned, the detection of dim, high-speed RSOs is an already challenging task that is made even more difficult by conditions such as daylight, moonlight, and atmospheric turbulence. Traditionally, small targets are detected using specialized radar or laser equipment, though ground-based optics have become an attractive alternative due to their power efficiency and cost effectiveness. However, optical charge-coupled device (CCD) sensors often struggle with high amounts of background noise as well as with long exposure times that can complicate the detection of fast-moving objects (Kong et al. (2019)). As an alternative optical device, event-based cameras could be ideal for replacing or complementing conventional CCD sensors for RSO detection. Recent work has already shown the use of event-based cameras for daytime imaging of LEOs (Cohen et al. (2019)), and simulated work has investigated star tracking using event-based data (Chin et al. (2019)). These successes, in addition to the successful application of object detection models such as YOLOv3 on space imaging datasets (Fletcher et al. (2019)), have motivated our work in investigating space object detection with event-based cameras. Furthermore, recent advances in space scene simulation have improved the ability to experiment with high fidelity, optical space collections. In this work, we make use of the *SatSim* simulator to generate the large number of samples necessary for model training (Cabello & Fletcher (2022)).

## 2.2 Event-based Classification and Object Detection

Due to the asynchronous nature of event-based data, ordinary computer vision techniques are not readily applicable to performing classification or object detection with event-based cameras. In the simplest case, event streams can be accumulated over time into approximated images that can then be used with conventional algorithms, but this eliminates much of the rich, temporal information. Early methods of working with event-based data often relied on updating existing algorithms, such as the Harris corner detector (Ni et al. (2012)) or Hough transform (Vasco et al. (2016)), to be compatible with asynchronous data. More sophisticated approaches, such as the HOTS (Lagorce et al. (2014)) and HATS algorithms (Sironi et al. (2018)), use new representations of event-based data (time surfaces), to exploit temporal information for classification and object detection. Newer approaches have begun to modify established machine learning models, such as YOLE or "You Only Look at Events", which adapts the YOLO object detection framework for asynchronous operation (Cannici et al. (2019)). However, one of the chief challenges for algorithm development and model training is the relative lack of event-based data publicly available. While neuromorphic versions of well-known datasets such as N-MNIST, CIFAR10-DVS, and N-Caltech101 exist, all of these datasets are generated using event-based extrapolations of the original datasets rather than actual event-based samples (Li et al. (2017)) (Orchard et al. (2015)). However, the N-Cars and Prophesee Gen1-Automotive datasets are two more recent datasets that include real event-based data collections (Sironi et al. (2018))(de Tournemire et al. (2020)).

## 2.3 Spiking Neural Networks

Given the asynchronous nature of event-based data, SNNs are a natural choice for performing classification and object detection with event-based cameras. Multiple works have already demonstrated effective use of SNNs for performing a wide array of tasks on event-based data, not only limited to classification and object detection (Lee et al. (2020))(Samadzadeh et al. (2020)). Furthermore, the innate compatibility with SNNS have made event-based camera datasets such as CIFAR10-DVS useful for evaluating a range of spiking models and encoding processes (Fang et al. (2021))(Vicente-Sola et al. (2021)). However, the greatest challenge to employing SNNs is the method of training used. The behavior of spiking neurons is not differentiable and therefore not immediately trainable using ordinary backpropagation. Also, as previously mentioned, the vanishing spike phenomenon is a further limiting factor on the potential depth of SNN models. As a result, a great deal of research has gone into finding new methods of spike-based backpropagation, surrogate gradients, or entirely new training methods (Bellec et al. (2018))(Wu et al. (2018))(Huh & Sejnowski (2018)). Some of the best results in terms of model accuracy have come from converting pre-trained artificial neural networks (ANNs) into spiking models, but this method can incur losses in the efficiency and speed of the resulting SNNs (Rathi et al. (2020)). Despite many possible solutions, training methods for SNNs continues to be an area of great interest.

## 3 Methods

In the following section, we describe the weighted spike representation used to encode event-based vision streams for use with deep spiking neural networks. We also detail the integration of this encoding process with the spiking neural networks and the truncated, surrogate gradient based learning used to train the large networks used. Finally, we describe the method by which optimal spiking hyperparameters are chosen to maximize both classification and object detection performance.

## 3.1 Weighted Spike Encoding

Given a dynamic vision sensor of width and height $W \times H$, the stream of events generated by such a sensor would be of the form

$$E_N = e_n|_{n=1}^N, e_n = (x_n, y_n, t_n, p_n) \tag{1}$$

where $E_N$ represents the entire stream of $N$ events and each event $e_n$ is of the form $(x_n, y_n, t_n, p_n)$. In this context, $x_n$ and $y_n$ are in the range $[1, W]$ and $[1, H]$ respectively, and represent the pixel location at which the event occurred, while $t_n$ is the timestamp associated with the generated event.

$p_n$ is the polarity of the event with a value of $p_n = \in \{-1, 1\}$ indicating that the event is generated by either a increase of luminance, +1, or decrease in luminance, -1. In order to use conventional computer vision techniques with event-based data, an event stream can be integrated over either a particular number of events $N$ or a chosen range of time $\Delta t$ with events accumulated at their respective $(x, y)$ locations.

$$I = \begin{cases} I_{[0,W,H]} = \sum_{n=0}^{N} \delta(x_n, y_n) & p_n = +1 \\ I_{[1,W,H]} = \sum_{n=0}^{N} \delta(x_n, y_n) & p_n = -1 \end{cases} \forall e_n \tag{2}$$

In general, event streams are integrated as described in Eqn. 2, where timestamps are ignored and simple counts of events are accumulated at their respective locations $(x_n, y_n)$, with positive and negative polarity events separated into separate channels to produce $2 \times W \times H$ images suitable for image-based models and conventional algorithms. In this work, we use this integration in conjunction with standard object detection and classification models as a baseline for performance comparison.

In order to reduce the possible number of spikes in dense event-based data, while also encoding some of the temporal information into the resulting output, we employ a modified form of event stream integration. While event-based data can be used natively with asynchronous input spiking neural networks, our encoding greatly reduces the timescale over which spikes are presented to the end network and enables the application of conventional convolutional networks through additional pre-processing steps. Our approach to encoding involves a partial integration process that is dependent on whether feature extraction is performed in real-time or on previously recorded event streams. In the real-time case, we choose a time window, $\Delta t$, from which events will be accumulated over the event stream until a total of $T$ timesteps, or windows, have been presented to the network. Conversely, with previously recorded event streams, we can choose the number of timesteps, $T$, and then determine the time window, $\Delta t$, required to evenly divide the event stream into the desired number of timesteps.

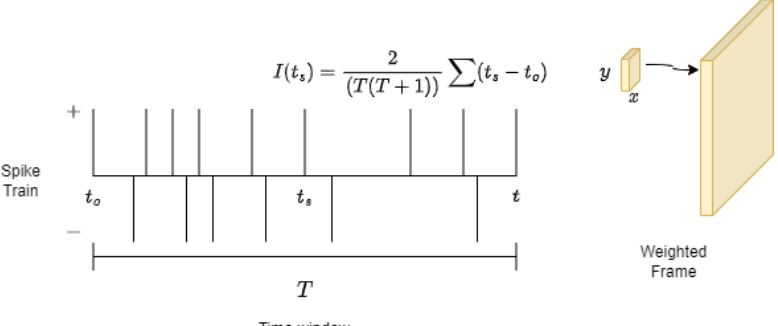

Figure 1: Generation of a temporally-weighted spike frame from an event stream.

Figure 1 depicts an event stream with a window of events accumulated over a given time range, $\Delta t$. In order to capture temporal information in a reduced representation, events at each x-y location added as the time difference of their corresponding timestamp with that of the first timestamp in the associated window, i.e. $t_s - t_o$. Once integrated, temporal weights are normalized over the entire range of possible timestamps within the window and organized into a two-dimensional frame according to their x/y locations. It is also important to note that this integration encoding is applied separately to positive and negative polarity events such that the final frames have two channels similar to ordinary integrated frames. This encoding is formalized in Eqn. 3.

$$I(p_n, W, H) = \begin{cases} I_{[0,W,H]} = \frac{2}{\Delta t(\Delta t+1)} [\sum_{n=0}^{N} \delta(x_n, y_n) * (t_n - t_o)] & p_n = +1 \\ I_{[1,W,H]} = \frac{2}{\Delta t(\Delta t+1)} [\sum_{n=0}^{N} \delta(x_n, y_n) * (t_n - t_o)] & p_n = -1 \end{cases} \forall e_n \tag{3}$$

Once encoded, the new sequence of temporal weight frames can be used either directly such as in a conventional 3D convolutional network or as modified spikes as in our work. Alternatively, the

frames can be unrolled into individual elements with thresholds applied to produce binary spikes for use with non-convolutional spiking neural networks; however, the effectiveness of this approach is not explored in this work.

## 3.2 SPIKING NEURAL NETWORK INTEGRATION

For performing the classification and object detection, we explore two avenues for incorporating spiking event data: a true spiking approach and a pseudo-spiking approach. For our true spiking approach, we use the discretized version of the leaky integrate and fire neuron proposed in Rathi et al. (2020). Each layer has an associated membrane potential, $u(t)$, that accumulates with the input of weighted spikes and generates spikes of its own if potential exceeds a voltage threshold, $u^t \geq V_t$. The dynamics are detailed in equations 4 and 5

$$u_i^t = \lambda u_i^{t-1} + \sum_j w_{ij} o_j^t - V_t o^{t-1} \tag{4}$$

$$o_i^{t-1} = \begin{cases} 1, u_i^{t-1} > V_t \\ 0, otherwise \end{cases} \tag{5}$$

where $t$ is the current timestep, $\lambda$ is a potential leak constant, $w_ij$ are weights associated with the previous layer, $o_j^t$ are outputs of previous layers, and $v$ is the voltage threshold. Eqn. 4 holds for the soft reset case, in which membrane potential is reduced by the voltage threshold upon firing. Empirically, we found better classification performance on deep networks when using the hard reset alternative, where membrane potential $u_i^t$ is reduced to 0 when $u^t \geq V_t$. We use a similar training process as in (Rathi et al. (2020)) by using the membrane potential at the final timestep as the network output needed to calculate the relevant loss metrics: cross-entropy in the classification case and the YOLO loss metrics (box regression, classification, and objectness) in the object detection case. However, since the exact timing of input spikes is altered due to our temporally weighted spike encoding, we opt to use the surrogate gradient

$$\frac{\delta o}{\delta u} = \alpha max\{0, 1 - |u - V_t|\} \tag{6}$$

where $o$ is the layer output, $u$ is membrane potential, and $V_t$ is the voltage threshold for output generation. This surrogate gradient for the spiking neuron output is then used with backpropagation through time (BPTT) in order to perform supervised training with spatial and temporal credit assignment.

Conversely, we posit that, for our purposes, a pseudo-spiking approach that removes the need for temporal credit assignment could still benefit from the temporal information found in the encoded event stream. Removing the need for BPTT should enable much faster convergence and potentially the effective training of much larger models normally too memory intensive for true spiking neural networks. In order to remove the need for temporal assignment, we present the entire spike train to each layer in series, such that membrane potential is accumulated for all timesteps $T$ on each layer before passing output to the succeeding layer. In order to impart some of the temporal information in this pseudo-spiking format, we use the membrane potential directly as the output of active layers. Equation 7 describes the process for the initial layer, which is also depicted fully in Figure 2, where Equations 8 and 9 detail the behavior for all successive layers in the network.

$$U(T) = \sum_T \lambda * I(t_i) \tag{7}$$

$$U_i(T) = \sum_T [\lambda U_i(t) - V_t O(t-1)] + \sum_j w_i j O_j(T) \tag{8}$$

$$O_j^T = \begin{cases} U_j(T), U_j(T) > V_t \\ 0, otherwise \end{cases} \tag{9}$$

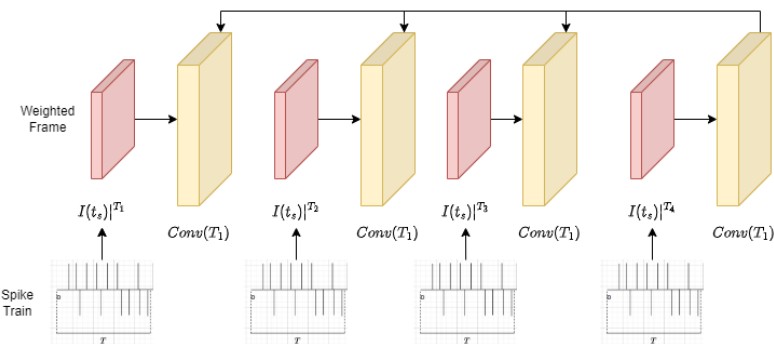

Figure 2: Accumulation of temporally-weighted spikes on the membrane potential of a pseudo-spiking convolutional layer.

As previously mentioned, the positive and negative polarity events (or ON and OFF events, respectively) of the input stream are encoded separately. After encoding, the positive and negative spikes for a particular time window are accumulated into two-dimensional frames and concatenated to form two-channel spike frames with shape similar to that of an ordinary image. Figure 3 depicts the broad process applied to a sample of the N-MNIST dataset.

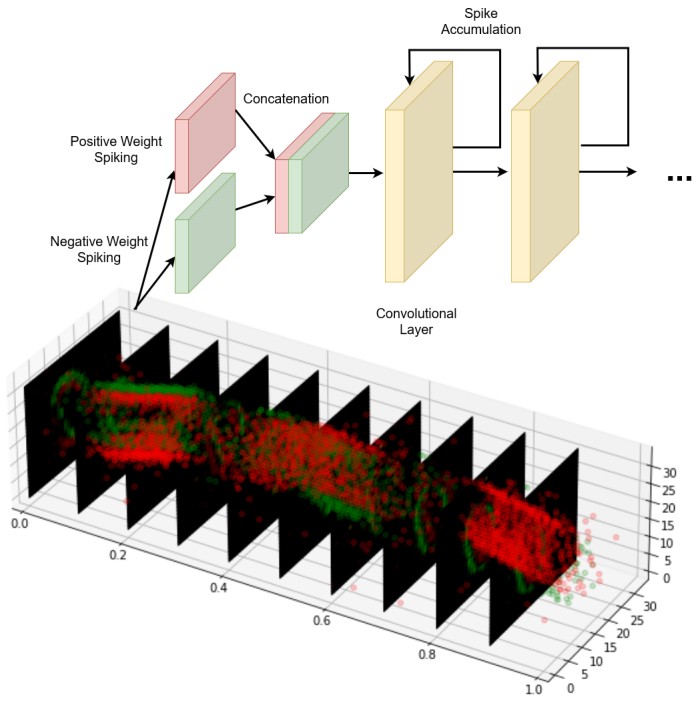

Figure 3: N-MNIST sample encoded and passed to pseudo-spiking convolutional network.

## 4    EXPERIMENTS

### 4.1    SPIKING HYPERPARAMETER SEARCH

In order to choose optimal hyperparameters necessary for spiking behavior, we use the hyperparameter optimization package *Optuna* (Akiba et al. (2019)). We perform hyperparameter optimization across three network architectures: VGG19, ResNet50, and DarkNet53. While VGG19 and

ResNet50 are the feature extraction backbones used for classification tasks, we also optimize hyperparameters for DarkNet53 to increase object detection performance and to investigate the efficacy of training larger networks with the pseudo-spiking behavior and encoding. Each feature extraction backbone is trained on a classification task, using the CIFAR10-DVS dataset, for 10 epochs with hyperparameter search performed for voltage threshold ($V_t$), leakage ($\lambda$), gradient output scaling ($\alpha$), and total timesteps ($T$). Hyperparameters are optimized solely for classification accuracy, which will tend to maximize timesteps for the pure spiking case due to the strong positive correlation between spiking timesteps and accuracy. Highest performing parameters are shown in Table 1.

Table 1: Optimal Spiking Hyperparameters

| Architecture | Voltage Thresh. ($V_t$) | Leakage ($\lambda$) | Gradient Scale ($\alpha$) | Timesteps (T) |
|---|---|---|---|---|
| sVGG19 | 3.191 | 0.325 | 2.74 | 185 |
| sResNet50 | 2.475 | 0.127 | 1.73 | 113 |
| pseudo-VGG19 | 2.002 | 0.145 | 5.58 | 12 |
| pseudo-ResNet50 | 2.505 | 0.07 | 3.79 | 39 |
| pseudo-DarkNet53 | 1.622 | 0.054 | 6.51 | 95 |

Based on the remaining hyperparameter tuning results, we find that the parameters that correlate most strongly with model size are both the leakage coefficient and the number of spiking timesteps. In general, larger spiking models requiring a greater number of timesteps in order to reach maximum accuracy is expected, and aligns with other previous results with SNN training. However, the leakage coefficient being exceptionally small for both ResNet50 and DarkNet53 suggests accuracy is maximized when output is generated at nearly every timestep for these larger models. This result may be explained by the overall reduction of input spikes due to the temporal encoding driving the need for neurons to fire more consistently.

## 4.2 CLASSIFICATION TASKS

While the primary focus of our work is to adapt event streams and spiking networks for object detection, we also include performance on classification tasks as both a means of verification and as a point of comparison with other methods of performing classification on neuromorphic data. Classification results were evaluated for the N-MNIST, CIFAR10-DVS, N-Cars, and N-Caltech101 datasets and compared to some of the most prominent results from literature. Although not widely used as an event-based classification dataset, we chose to include the N-Cars classification dataset due to its relation to the GEN1-Automotive dataset used more prominently for object detection. In all instances, the indicated model architectures for our method use the same layer structure as available in pretrained, PyTorch model zoo (Paszke et al. (2019)), albeit with convolutional layer behavior replaced by the pseudo-spiking behavior previously described.

Table 2: Classification Accuracies (Top 2 best classification accuracies per dataset are bolded).

| Method | N-MNIST | CIFAR10-DVS | N-Cars | N-Caltech101 |
|---|---|---|---|---|
| sVGG19 + Encoding | 0.975 | 0.7151 | 0.911 | 0.689 |
| sResNet50 + Encoding | 0.898 | 0.6119 | 0.845 | 0.615 |
| pseudo-VGG19 + Encoding | **0.997** | **0.7404** | 0.924 | **0.728** |
| pseudo-ResNet50 + Encoding | 0.974 | 0.6510 | 0.889 | 0.662 |
| Sironi et al. (2018) | 0.991 | 0.524 | 0.902 | 0.642 |
| Cannici et al. (2019) | - | - | **0.927** | 0.702 |
| Messikommer et al. (2020) | - | - | **0.944** | **0.745** |
| Wu et al. (2019) | 0.9953 | 0.605 | - | - |
| Samadzadeh et al. (2020) | **0.996** | 0.692 | - | - |
| Vicente-Sola et al. (2021) | - | 0.7298 | - | - |
| Fang et al. (2021) | **0.996** | **0.748** | - | - |

In terms of classification accuracy across the general datasets, we see that the VGG19 architecture outperforms the deeper ResNet50 models, but both pseudo-spiking architectures generally outper-

form their fully spiking counterparts. As intended, the performance gap between the pseudo-spiking and fully spiking methods is greater for the ResNet50 model, suggesting that the pseudo-spiking behavior and temporally-weighted encoding has a positive impact on the training of deeper architectures.

## 4.3 OBJECT DETECTION TASKS

For our primary SDA task, we have two object detection datasets denoted *Satellite Collect* and *SatSim* in Table 3. As previously mentioned, the *SatSim* space scene simulator is capable of generating high-fidelity, event-based simulations of satellite collections with a degree of tunable conditions and sensor parameters. Given the small size of real data collections available (about 900 samples) in the *Satellite Collect* dataset, we first generate a large collection of simulated samples with approximately equivalent parameters using the *SatSim* simulator. As a basis for showing improvement, we compare results from the pseudo-spiking YOLO model with temporally-weighted encoding to both a conventional YOLO model using an integrated frame and a pseudo-spiking YOLO model using spikes generated from Poisson-encoded integrated images. Poisson-encoding has been used in multiple previous works for generating spike frames from ordinary images (Rathi et al. (2020)), and we use this encoding here to gauge the effectiveness of adding the temporally-weighted encoding. Both spiking and non-spiking models are trained and evaluated on this dataset, after which all models are retrained and evaluated on the real dataset to best estimate real-world performance. In the context of the SDA datasets, each dataset has only a single class (satellite) and the goal is to achieve maximal detection. As a result, the metric of greatest interest for the SDA task is the maximum $F_1$ score, and full precision-recall results are shown for these datasets only. Figure 4 shows an example of real optical, event-based satellite collection and a simulated approximation generated by the *SatSim* simulator.

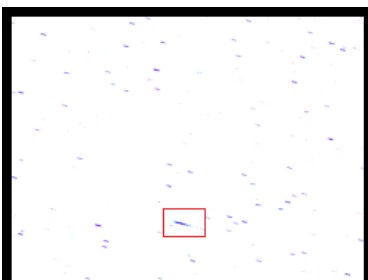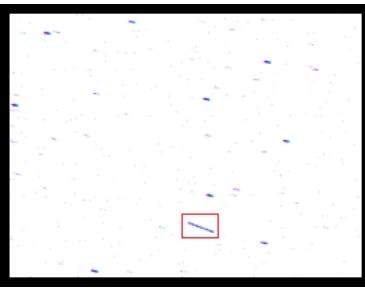

Figure 4: Real satellite collection (left) vs. SatSim generated sample (right). Samples are generated as full event streams, but displayed here as integrated frames.

For general comparison, we also evaluate our method on two publicly available datasets: N-Caltech101 and the Prophesee GEN1-Automotive dataset. As is customary for general object detection, we show the mean average precision (mAP) (Everingham et al. (2010)) results for our method and those of other state-of-the-art asynchronous detection methods.

Table 3: Performance results for spiking object detection models and conventional models on equivalent datasets.

| Method | Dataset | Precision | Recall | $F_1^*$ | mAP |
|---|---|---|---|---|---|
| YOLO+Int. Frame | Satellite Collect | 0.43909 | 0.65570 | 0.52597 | - |
| pseudo-sYOLO+Int. Frame | Satellite Collect | 0.61765 | 0.88489 | 0.72750 | - |
| pseudo-sYOLO+Enc. | Satellite Collect | 0.67804 | 0.90103 | **0.7740** | - |
| YOLO+Int. Frame | SatSim | 0.65179 | 0.65710 | 0.65443 | - |
| pseudo-sYOLO+Int. Frame | SatSim | 0.745332 | 0.739609 | 0.74246 | - |
| pseudo-sYOLO+Enc. | SatSim | 0.77709 | 0.74871 | **0.76264** | - |
| pseudo-sYOLO+Int. Frame | N-Caltech101 | - | - | - | 0.331 |
| pseudo-sYOLO+Enc. | N-Caltech101 | - | - | - | **0.595** |
| Cannici et al. (2019) | N-Caltech101 | - | - | - | 0.398 |
| Messikommer et al. (2020) | N-Caltech101 | - | - | - | **0.643** |
| pseudo-sYOLO+Int. Frame | GEN1-Auto | - | - | - | 0.124 |
| pseudo-sYOLO+Enc. | GEN1-Auto | - | - | - | **0.339** |
| Messikommer et al. (2020) | GEN1-Auto. | - | - | - | 0.149 |
| Cannici et al. (2020) | GEN1-Auto. | - | - | - | 0.31 |
| Perot et al. (2020) | GEN1-Auto. | - | - | - | **0.40** |

Across these object detection results, we see a significant increase in the maximum $F_1$ score for both real and simulated satellite data collections when using a pseudo-spiking YOLO as compared to a conventional YOLO model on integrated frames. Furthermore, the pseudo-spiking model using the temporally-weighted encoding displays another significant increase in performance that appears to be even more pronounced on real collections over the simulated equivalents. This may be a result of the less uniform timestamps of generated events in real versus simulated data, which suggests that the temporally-weighted encoding is successfully preserving temporal data lost in the integrated frames. In terms of the public datasets, the pseudo-spiking YOLO model with temporally-weighted encoding shows competitive mean average-precision (mAP) on N-Caltech101 and the Prophesee GEN1-Automotive dataset, with its performance only beaten out by some of the most recent and more complex asynchronous methods available.

## 5 CONCLUSION

In this work, we have presented a new temporally-weighted spike encoding for event-based camera streams that greatly reduces the number of spikes required for processing noisy event streams, while preserving useful temporal information. We have additionally demonstrated a pseudo-spiking behavior for convolutional layers that allows us to mimic properties of a spiking network, but allows us to train deep networks with conventional backpropagation. Both object detection and classification results show that the combination of temporally-weighted spike encoding and pseudo-spiking behavior increase accuracy and performance, especially when used on deeper models. For our own area of interest, this method also demonstrates superior performance on object detection tasks for space-domain awareness, while also generalizing well to publicly available datasets.

In the future, we hope to assess the training of a larger array of models, as well as incorporate the method with newer versions of object detection models. Furthermore, we have introduced the method by which the temporally-weighted spike encoding could be used to process event streams in real-time, but this potential is as of yet unexplored. Real-time object detection warrants an exhaustive study into the potential energy and memory-saving benefits, and could also highlight the comparative strengths of event-based cameras for SDA tasks.

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
