# OpenReview forum: "Temporally-Weighted Spike Encoding for Event-based Object Detection and Classification"
_ICLR.cc/2023/Conference — Submitted to ICLR 2023_

### Official Review · Reviewer_Uhms · 2022-10-21

**Confidence:** 3
**Correctness:** 2
**Technical Novelty And Significance:** 2
**Empirical Novelty And Significance:** 2
**Recommendation:** 3

**Clarity, Quality, Novelty And Reproducibility:**

Overall I do not feel there is a lot of novelty in the paper and the claim that this leads to major spiking efficiencies is not supported enough. Unless I missed it somewhere, I do not see a mention that source code will be provided.

**Strength And Weaknesses:**

Strengths:

S1: papers on SNN are always welcome as they hold the promise of efficient low power inference

Weaknesses:

W1: why do the authors limit themselves to DVS datasets which are not as popular as the standard RGB datasets? There exists a significant literature on SNN tested with RGB datasets and achieving close to state of the art performance on this RGB data. Typically this is accomplished by having a transduction layer (typically a CNN) as the first layer that converts the RGB data to spike data. Note that often a temporal window of 1 is used (ie up to 1 spike per pixel is output per frame). See for example "Convolutional networks for fast, energy-efficient neuromorphic computing" Esser et al 2016 for an example demonstrating SOA performance (many more similar papers since then).

W2: use \times instead of x to denote cross product

W3: page 4, eq 3: why is this done with respect to t_0 and not with respect to the median timestamp for example. Overall spend more time explaining what the advantage of this approach is

W4: eq 4, eq 5: should j be a subscript?

W5: each layer seems to output a rate code and the algorithm accumulates. This does not seem really efficient power wise or time wise. There exist much more efficient algorithms in the literature as I alluded to above. Overall the argument that the number of spikes is reduced significantly does not seem to be supported enough. Quantify this better. Put a table with the average and maximum number of spikes per frame achieved by this algorithm vs the competition

W6: Section 4.3 "Figure ??". Missing figure number

**Summary Of The Paper:**

The authors present a spike encoding scheme that they argue reduces the number of spikes from an event based spike stream. They argue this enables the training of larger SNNs with fewer timesteps.

They use this representation to train an SNN and test their trained model on DVS datasets such as N-MNIST and DVS-CIFAR10 and N-Caltech.

**Summary Of The Review:**

Overall for the reasons I mentioned above I do not feel this paper is at the level of ICLR

---

### Official Review · Reviewer_AfjB · 2022-10-23

**Confidence:** 5
**Clarity, Quality, Novelty And Reproducibility:** 1)The authors should carefully check …
**Correctness:** 2
**Technical Novelty And Significance:** 1
**Empirical Novelty And Significance:** 1
**Recommendation:** 3

**Strength And Weaknesses:**

(1)Strength:

The explored issue is challenging but intriguing.

(2)Weakness:

1) In Sec 3.1, the authors state that “In order to preserve temporal information, .... employ a modified form of event stream integration.”  Nonetheless, the reviewer thinks that any temporal integration of event data would inevitably degrade the original temporal data. Consequently, the authors should revise their motivations or alter the network inputs.

2) Applying the spiking neural network [1,2] to the frame-like representation of event data has been widely adopted for event data processing. The novelty of the proposed method is very limited.

3) The authors should note the best or second-best approaches in Tables 2 and 3.

4) The authors should carefully check the format of the paper. The captions should be placed above tables. The authors should revise the citation for Figure 4. In Tables 1 and 2, the bottom bounds are absent. Table 3 is beyond the limited scope.

5) It is quite difficult to determine the specific details of Figure 4. Authors should adopt other representations.

6) The authors should compare the pseudo-spiking layer with the spiking layer for the item recognition task in table 3.

7) Authors should compare the computational complexity of different methods, such as FLOPs, number of parameters, and execution time.

8) The authors must redraw Figures 1 and 2. Currently, it is difficult to understand them.

9) For the object identification studies, the authors only compared with a very basic baseline, YOLO + Int. Frame. Numerous effective event-based approaches [3] or event-branch fusion-based detectors [4,5] exist. Authors should compare with them.

[1] Li J, Dong S, Yu Z, et al. Event-based vision enhanced: A joint detection framework in autonomous driving[C]//2019 ieee international conference on multimedia and expo (icme). IEEE, 2019: 1396-1401.

[2] Zhang J, Dong B, Zhang H, et al. Spiking Transformers for Event-Based Single Object Tracking[C]//Proceedings of the IEEE/CVF Conference on Computer Vision and Pattern Recognition. 2022: 8801-8810.

[3] Li J, Li J, Zhu L, et al. Asynchronous Spatio-Temporal Memory Network for Continuous Event-Based Object Detection[J]. IEEE Transactions on Image Processing, 2022, 31: 2975-2987.

[4]Liu M, Qi N, Shi Y, et al. An attention fusion network for event-based vehicle object detection[C]//2021 IEEE International Conference on Image Processing (ICIP). IEEE, 2021: 3363-3367.

[5] Tomy A, Paigwar A, Mann K S, et al. Fusing Event-based and RGB camera for Robust Object Detection in Adverse Conditions[C]//2022 International Conference on Robotics and Automation (ICRA). IEEE, 2022: 933-939.


**Summary Of The Paper:**

This paper investigates the problem of event-based object detection and classification. The authors introduce a temporal integration-based event representation. Then, a modified pseudo-spiking neural network is applied to extract event representations for object classification and detection. However, the novelty of the proposed method is limited. Additionally, the performance is unsatisfactory.

**Summary Of The Review:**

The novelty of the proposed method is limited, and the performance is unsatisfactory. The writing and organization should be significantly improved. Therefore, the submission is far away from being published.

---

### Official Review · Reviewer_gpjB · 2022-10-24

**Confidence:** 5
**Correctness:** 3
**Technical Novelty And Significance:** 3
**Empirical Novelty And Significance:** 3
**Recommendation:** 6

**Clarity, Quality, Novelty And Reproducibility:**

The work is in general clearly written but would benefit from a novel checking of syntax. Some minor points:
 • "can results" in the abstract
 • Use the latex `\times` for multiplying dimensions as in `W x H` above eq (1)
 • "Figure??" page 8
 More generally, the paper would benefit in general to better highlight the novelty of your contributions with respect to SOTA not only in terms of accuracy, but in the interpretation of the resulting networks.

**Strength And Weaknesses:**

Equation (3) is the core of the novelty brought by the paper, yet it is at first difficult to understand. First, it uses a different convention as the standard spike accumulating described in Equation (2) Then you should specifically define the events $e_n$ for which you accumulate information. From Equations (7-9), it is similar to the HOTS approach, yet with a different kernel and with events within the time window (instead of the time to last spike). It would therefore be important to justify that choice and compare the accuracy of your model with HOTS for instance. Also, given the leakage term (Eq 7) how do you interpret your optimisation in table 1 with respect to the different datasets? Why not use different constants at different layers?

**Summary Of The Paper:**

Authors propose to train spiking neural networks using a temporally-weighted spike encoding. This is tested on standard backbones with a classical surrogate gradient learning. It is then applied on several benchmarks, but also on an interesting star-tracking application.


**Summary Of The Review:**

The paper proposes a novel encoding of event-based inputs which is applied to generic benchmark but also to a star-hacking application. It shows promising results, yet relatively few qualitatives results are shown to understand why this encoding would be better than others.

Do you observe different results with different parameterisations of your encoding? Do you observe more explainable kennels? Also, the star-tracking applications seems like an "easier" one. Could you a similar result with a shallower network?

---

### Official Review · Reviewer_WXPN · 2022-10-26

**Confidence:** 3
**Correctness:** 3
**Technical Novelty And Significance:** 2
**Empirical Novelty And Significance:** 2
**Recommendation:** 6

**Clarity, Quality, Novelty And Reproducibility:**

The encoding is essentially an integrated time window of events, which is explored in previous works. The novelty lies in the encoding and using this representation with spike neurons.

Overall, I think the paper could be strengthened by reducing the focus on the spiking neuron description and integration, and emphasizing the novelty of the representation itself (as the main focus of the work); a more thorough evaluation against competing encoding schemes and, possibly, more challenging classification datasets would be beneficial - although I acknowledge that this might be difficult.


Typo in Abstract: "can results in"

**Strength And Weaknesses:**

The spike-based learning is a very relevant but often overlooked field in event-based vision; as the paper presents a method with a competitive accuracy scores and a potential to improve computational performance, this encoding scheme could potentially have a good impact within the spike-based learning community.

One of the benefits presented was the possibility to reduce the computational cost, but the evaluation section omits the results on the computational performance. It would be useful to include a holistic evaluation of training and inference computational cost, and hyperparameter (such as temporal window size) impact.

**Summary Of The Paper:**

A novel spike encoding system for event streams is presented, that allows to reduce the amount of spikes and increase the performance of spike-learninig event-based pipelines. The method is evaluated on recognition benchmarks, as well as on Earth satellite tracking datasets.

**Summary Of The Review:**

I believe the paper has good potential for publication, but could benefit from some improvements to the text structure (emphasizing the contributions more), and adding the computational performance evaluation.

---

### Decision · Program_Chairs · 2023-01-20

**Decision:**

Reject

**Justification For Why Not Higher Score:**

Seems evident that also the above borderline reviews triggered some concerns about experimental validation and clarity. Also they appear to be lighter and therefore I am inclined to weight more the low scores in this case.

**Justification For Why Not Lower Score:**

N/A

**Metareview: Summary, Strengths And Weaknesses:**

The paper presents a temporally-weighted spike encoding that aims at greatly reducing the amount of spikes and increase the performance of spike-learninig event-based pipelines with SNN. The method is evaluated on recognition benchmarks, as well as on Earth satellite tracking datasets.
All authors agree that the work is novel, although some tend to consider the novelty rather marginal, there seem to be consensus on the need for more thorough experimental validation and analysis of results. A more rigorous performance evaluation is needed as well considering that this is the main objective of the proposed work. Clarity is also to be improved.
Therefore, I am afraid that the work is not ready for publication.